# Prognostic biomarkers for predicting papillary thyroid carcinoma patients at high risk using nine genes of apoptotic pathway

**Chakit Arora, Dilraj Kaur, Leimarembi Devi Naorem, Gajendra P. S. Raghava** [ORCID] *

Indraprastha Institute of Information Technology-Delhi, Department of Computational Biology, New Delhi, India

* raghava@iiitd.ac.in

**Data Availability Statement:** The data used in this study is available from the TCGA-GDC portal at

## Abstract

Aberrant expressions of apoptotic genes have been associated with papillary thyroid carcinoma (PTC) in the past, however, their prognostic role and utility as biomarkers remains poorly understood. In this study, we analysed 505 PTC patients by employing Cox-PH regression techniques, prognostic index models and machine learning methods to elucidate the relationship between overall survival (OS) of PTC patients and 165 apoptosis related genes. It was observed that nine genes (*ANXA1*, *TGFBR3*, *CLU*, *PSEN1*, *TNFRSF12A*, *GPX4*, *TIMP3*, *LEF1*, *BNIP3L*) showed significant association with OS of PTC patients. Five out of nine genes were found to be positively correlated with OS of the patients, while the remaining four genes were negatively correlated. These genes were used for developing risk prediction models, which can be utilized to classify patients with a higher risk of death from the patients which have a good prognosis. Our voting-based model achieved highest performance (HR = 41.59, p = $3.36 \times 10^{-4}$, C = 0.84, logrank-p = $3.8 \times 10^{-8}$). The performance of voting-based model improved significantly when we used the age of patients with prognostic biomarker genes and achieved HR = 57.04 with p = $10^{-4}$ (C = 0.88, logrank-p = $1.44 \times 10^{-9}$). We also developed classification models that can classify high risk patients (survival ≤ 6 years) and low risk patients (survival > 6 years). Our best model achieved AUROC of 0.92. Further, the expression pattern of the prognostic genes was verified at mRNA level, which showed their differential expression between normal and PTC samples. Also, the immunostaining results from HPA validated these findings. Since these genes can also be used as potential therapeutic targets in PTC, we also identified potential drug molecules which could modulate their expression profile. The study briefly revealed the key prognostic biomarker genes in the apoptotic pathway whose altered expression is associated with PTC progression and aggressiveness. In addition to this, risk assessment models proposed here can help in efficient management of PTC patients.

## Introduction

Thyroid cancer's incidence has been reported to be increasing every year, having the fastest growth rate amongst all the cancers [1]. Thyroid cancer developed from follicular cells can be

https://portal.gdc.cancer.gov with the project name
TCGA-THCA.

**Funding:** The author(s) received no specific
funding for this work.

**Competing interests:** The authors have declared
that no competing interests exist.

mainly categorized into papillary (PTC), follicular (FTC), and anaplastic thyroid cancer
(ATC). PTC is the most common malignant subtype comprising about 80–85% of all thyroid
cancer incidences [2]. Although, it is associated with a good prognosis, around 20–30% of the
patients are reported to exhibit poor prognosis. This is mostly attributed to the development of
distant tumour metastases and recurrences. The progression/transformation of PTC to a more
aggressive state, i.e. a poorly differentiated state or a non-differentiated state such as ATC has
also been observed in some cases. Thus, efficient risk stratification methods are required for
prognostic evaluation and therapeutic decision making in PTC patients. Conventional risk
stratifications rely on clinico-pathological factors such as age, gender, tumour size, tumour
spread and stage [3, 4] but these are plagued with limitations and uncertainties. These limita-
tions demand novel risk assessment methods which are more accurate and derivable from the
primary mechanisms driving PTC oncogenesis.

Due to advent of high-throughput sequencing methods and public databases, many biomark-
ers have been identified for PTC diagnosis, classification, and prognosis. These biomarkers, are
important for understanding molecular mechanisms of thyroid cancer. Classic examples include
*BRAF* mutation status, *RET/PTC* and *PAX8/PPAR* rearrangements [5–7]. *BRAF* mutations at
*V599E* and *T1799A* are known to induce the serine kinase levels and thus activate *MAPK* path-
way. Similarly, *RET/PTC* rearrangement regulate the *NFkB* activity and thus promote PTC cell
migration. Another example of rearrangement is *PAX8/PPAR* that mediates the transcription
pathway and advances PTC progression. In the past, several gene expression-based biomarker
have been reported that play a crucial role in PTC prognosis; owing to their altered/differential
expression profiles. For example *FOXF1* (HR:0.114, 95%CI: 0.045–0.289) and *FMO1* (HR:0.202;
95% CI: 0.084–0.487) genes were shown to be associated with favourable RFS (recurrence free
survival) in PTC patients [8, 9]. Downregulation of *FOXF1*, the gene belonging to the forkhead
family of TFs (transcription factors), was also seen to be related with advanced T staging, nodal
invasion, and late pathological staging. It has been observed that high expression of *FOXE1* a
member of forkhead family, also act as a tumour suppressor in PTC [10]. High expression of
*FOXE1* was found to negatively regulate *PDFGA* (target gene platelet-derived growth factor A)
expression in the early stage of PTC and thus affect the migration, proliferation and invasion of
PTC. Proteoglycans genes (e.g. *SDC1*, *SDC4*, *KLK7*, *KLK10*, *SLPI*, *GDF15*) were found to be
overexpressed in PTC samples [11]. Similarly, lower expression of *VHL* gene was shown to be
associated with aggressive PTC features and DFI (disease free interval, logrank-p = 0.007) [12].
*VHL* (von Hippel–Lindau) protein, by acting as a substrate recognition unit in a multiprotein
complex with E3 ubiquitin ligase activity, is involved in the degradation of the proteins such as
*HIF-α*. Whereas *HIF-α* regulates the levels of various angiogenic factors and is thus negatively
affected, resulting in a reduced angiogenesis. Bhalla et al. [13] reported 36 RNA transcripts
whose expression profiles were used to distinguish early and late-stage PTC patients (AUROC
0.73). In addition to above, number of candidate genes and biomarkers have been reported in
previous studies [14–16]. Despite tremendous progress in the field of prognostic biomarkers,
still it is far from perfection. There is a need to develop methods to identify key regulators of crit-
ical subcellular mechanisms that can serve as prognostic biomarkers.

One such vital mechanism is programmed cell death or apoptosis. Apoptosis is the process
for eliminating cells in multicellular organisms. The process of apoptosis is orchestrated by a
multitude of molecules (such as *p53*, *Bcl2* family, *TRAIL*, *FAS*) which respond to various inter-
cellular and extra-cellular stresses such as DNA damage, hypoxia etc. The activation of apopto-
tic pathway through various responsive arms drives a cascade of signalling events ultimately
leading to the activation of "Caspases" and eventually the demise of the cell. Dysregulation of
apoptosis is responsible for many diseases including cancer. Numerous studies have identified
key biomarkers linked with the cellular apoptosis. Charles EM *et al.* present the literature

related to the apoptotic molecules implicated as biomarkers in melanoma [17]. Another review provides extensive information related to apoptotic biomarkers such as *p53*, *Bcl2*, *Fas/FasL*, *TRAIL* in colorectal cancer [18]. Several other studies have also identified key molecules with prognostic roles in other cancers like gastric cancer [19, 20], breast cancer [21], lung cancer [22], bladder urothelial carcinoma [23], glioblastoma [24] and osteosarcoma [25]. Apoptosis has also been found to have a crucial role in carcinogenesis of thyroid cancer. Alterations in an increasing number of apoptotic molecules such as *p53*, *Bcl2*, *Bcl-XL*, *Bax*, *p73*, *Fas/FasL*, *PPARG*, *TGFb and NFKb* have been associated with thyroid cancer [26]. Since apoptotic resistance is mostly accounted for tumour proliferation and aggressiveness, apoptotic pathway has also emerged as a crucial target to develop anticancer treatments for thyroid tumours. For example, paclitaxel and manumycin are known to stimulate *p21* expression and induce apoptosis in ATC [27]. Lovastin inhibits protein geranylation of the *Rho* family and thus induces apoptosis in ATC [28]. *UCN-01* inhibits expression of *Bcl-2*, leading to apoptosis [29]. Since apoptosis in PTC is a complicated multistep process involving a number of genes, it remains poorly understood and needs to be further explored at a genetic level.

In this study, we exploited the mRNA expression data obtained from The Cancer Genome Atlas-Thyroid Carcinoma (TCGA-THCA) cohort and identified key apoptotic genes that are associated with PTC prognosis. We further constructed multiple risk stratification models using these genes and evaluated the potential of these models for prognosis using univariate and multivariate analyses, Kaplan Meier survival curves and other standard statistical tests. The nine-gene voting based model was found to perform the best and also stratified high risk clinical groups significantly. Finally, after a comprehensive prognostic comparison with other clinico-pathological factors, we developed a hybrid model which combines the expression profile of nine genes with 'Age' to predict High and Low risk PTC patients with high precision. Moreover, we further validated the expression patterns of the prognostic genes by GEPIA and HPA database respectively and verified their important biological processes. We also catalogued candidate small molecules that can modulate the expression of these genes and could be potentially employed in the efficient treatment of PTC patients.

## Materials and methods

### Dataset and pre-processing

The original dataset consisted of quantile normalized RNAseq expression values for 573 Thyroid Carcinoma (THCA) patients that were obtained from 'The Cancer Genome Atlas' (TCGA) using TCGA Assembler-2 [30]. This dataset, with the project name TCGA-THCA, was downloaded on 14th Oct, 2019. Out of which, information about overall survival (OS) time and censoring was available for 505 patients. The list of genes involved in the apoptotic pathway were taken from previous study [31]. Thus, the final dataset was reduced to 505 samples, using in-house python and R-scripts, constituting RNAseq values for 165 apoptotic genes. More details about clinical, pathological and demographic features corresponding to the final dataset are summarized in Table 1 in S1 File.

### Survival analysis

Hazard ratios (HR) and confidence intervals (95% CI) were evaluated to predict the risk of death related to high- and low-risk groups based on overall survival time of patients. These were stratified on the basis of appropriate cut-offs for various factors, using the univariate unadjusted Cox-Proportional Hazard (Cox-PH) regression models. Kaplan-Meier (KM) plots were used to compare survival curves of the risk groups. 'survival' and 'survminer' packages were used to perform survival analyses on the dataset. log-rank tests were used to estimate the

**Table 1. The table shows results of univariate cox regression with >median cut-off.**

|  | Gene | HR | p-val | C | %95 CI L | %95 CI U | Logrank-p |
|---|---|---|---|---|---|---|---|
| 1. | *ANXA1* | 0.14 | $2.82 \times 10^{-3}$ | 0.72 | 0.04 | 0.51 | $7.35 \times 10^{-4}$ |
| 2. | *TGFBR3* | 5.68 | $7.90 \times 10^{-3}$ | 0.62 | 1.58 | 20.49 | $2.82 \times 10^{-3}$ |
| 3. | *CLU* | 0.18 | $8.15 \times 10^{-3}$ | 0.53 | 0.05 | 0.64 | $2.92 \times 10^{-3}$ |
| 4. | *PSEN1* | 0.15 | $1.20 \times 10^{-2}$ | 0.71 | 0.03 | 0.66 | $2.38 \times 10^{-3}$ |
| 5. | *TNFRSF12A* | 0.25 | $1.57 \times 10^{-2}$ | 0.51 | 0.08 | 0.77 | $1.30 \times 10^{-2}$ |
| 6. | *GPX4* | 0.27 | $2.98 \times 10^{-2}$ | 0.62 | 0.09 | 0.88 | $2.09 \times 10^{-2}$ |
| 7. | *TIMP3* | 3.49 | $3.52 \times 10^{-2}$ | 0.68 | 1.09 | 11.18 | $2.53 \times 10^{-2}$ |
| 8. | *LEF1* | 3.36 | $4.10 \times 10^{-2}$ | 0.68 | 1.05 | 10.77 | $3.00 \times 10^{-2}$ |
| 9. | *BNIP3L* | 4.56 | $4.78 \times 10^{-2}$ | 0.68 | 1.01 | 20.46 | $2.05 \times 10^{-2}$ |

Genes with HR>1 are BPM while HR<1 are GPM.

statistical significance between the survival curves. Concordance index (C) was computed to measure the strength of predictive ability of the model [32–34]; p-values less than 0.05 were considered as significant. Multivariate survival analysis based on Cox regression was employed to compare the relationship between various covariates.

## Multiple gene-based models

**Machine learning based regression (MLR) models.** Various regression models from 'sklearn package in Python [35] were implemented to fit the gene expression values against the OS time. Regressors such as Linear, Ridge, Lasso, Lasso-Lars, Elastic-Net, Random-forest (RF) and K-nearest neighbours (KNN) were used. Five-fold cross-validation was used for training and validation studies, as done in previous studies [36–40]. All five test datasets were combined as 'predicted OS' and stratification was performed using it. Median cut-off was used to estimate HR, CI and p-values. Hyperparameter optimization and regularization was achieved using the in-built function 'GridsearchCV'. Model's performance is denoted using standard parameters viz. RMSE (root mean squared error) and MAE (mean absolute error).

**Prognostic index (PI).** For *n* genes, Prognostic Index (PI) is defined as:

$$PI = \beta_1 g_1 + \beta_2 g_2 + \ldots + \beta_n g_n$$

Where $g_i$ represent genes and $\alpha_i$ represent regression coefficients obtained from Cox univariate regression analysis as done in [38, 41–45]. Risk groups were stratified based on best PI cut-off estimated using cutp from 'survMisc' package in R. HR, p-values, C index were then evaluated using this cut-off.

**Gene voting based model.** Corresponding to an individual gene expression (median cut-off), a risk label 'High Risk' or 'Low Risk' was assigned to each patient. Thus, for *n* survival associated genes, every patient was denoted by a 'risk' vector of *n* risk labels. In gene voting based method, the patient is ultimately classified into one of the high/low risk categories based on the dominant 'label' (i.e. occurring more than at least n/2 times) in this vector. This is followed by evaluation of standard metrics [44].

## Prognostic gene signature validation by GEPIA tool and HPA database

The expression of the nine prognostic genes was further verified at the mRNA level by GEPIA [46] (Gene Expression Profiling Interactive Analysis), a web-based server, and the protein level using immunostaining data from The Human Protein Atlas (HPA) database [47].

### Enrichment analysis of the gene signature

The identified prognostic genes were uploaded to GOnet tool (https://tools.dicedatabase.org/GOnet/) [48] for gene ontology functional annotation against *Homo sapiens* with q-value threshold of < 0.05.

## Results

### Survival associated apoptotic genes

Cox-Proportional hazard models were used to find those apoptotic pathway genes that are related with PTC patient survival (Table 2 in S1 File). A univariate Cox-PH analysis revealed a total of 5 good prognostic marker (GPM) genes i.e the genes that are positively correlated with patient OS time and 4 bad prognostic marker (BPM) genes which are negatively correlated with OS time of the patients. GPM genes are *ANXA1*, *CLU*, *PSEN1*, *TNFRSF12A* and *GPX4* while BPM genes are *TGFBR3*, *TIMP3*, *LEF1* and *BNIP3L*. Table 1 shows the results for these genes along with the metrics associated with stratification of high/low risk patients at median cut-off. The precise molecular information about these 9 genes and PMIDs of the studies pertaining to their role in cancer, as obtained from GeneCards [49] and The Candidate Cancer Gene Database (CCGD) [50] respectively, is provided in Table 3 in S1 File. Table 4 in S1 File shows results of risk stratification performed using various previously suggested prognostic genes in PTC using cox univariate analysis in TCGA-THCA dataset at median expression cut-off for overall-survival.

### Risk estimation using multiple gene-based models

Several risk stratification models based on MLR, prognostic index and gene voting were constructed using the expression profile of nine survival associated apoptotic genes. Table 2 shows the results corresponding to various risk models. Amongst these, the performance of gene voting based model was found to be the best with HR = 41.59 and p~$10^{-4}$ with C-value of 0.84. In addition, high/low risk groups survival curves were significantly separated with a logrank-p~$10^{-8}$ using voting-based model. As shown in KM plot (Fig 1), 10-year survival rate for low risk patients was close to 98%, for high risk patients it was drop to 40%. PI based model performed the second best with HR = 17.55 and p~$10^{-3}$ (Fig 1 in S2 File), and regression-based RF model was the third best (and top amongst MLR models) with HR = 3.09 but p-value was found to be statistically insignificant.

**Table 2. The performance of different models developed using multiple gene expression profile-based method.**

| | Model | HR | p-val | C | %95 CI L | %95 CI U | Logrank-p |
|---|---|---|---|---|---|---|---|
| **1.** | **Voting based** | **41.59** | **3.36 x10$^{-4}$** | **0.84** | **5.42** | **319.17** | **3.80 x10$^{-8}$** |
| 2. | PI | 17.55 | 5.88 x10$^{-3}$ | 0.65 | 2.29 | 134.72 | 6.73 x10$^{-5}$ |
| 3. | RF | 3.09 | 8.43 x10$^{-2}$ | 0.68 | 0.86 | 11.09 | 5.91 x10$^{-2}$ |
| 4. | Linear | 1.59 | 3.98 x10$^{-1}$ | 0.54 | 0.54 | 4.65 | 4.04 x10$^{-1}$ |
| 5. | KNN | 1.09 | 8.68 x10$^{-1}$ | 0.56 | 0.38 | 3.12 | 8.68 x10$^{-1}$ |
| 6. | Lasso | 1.07 | 9.06 x10$^{-1}$ | 0.52 | 0.37 | 3.08 | 9.06 x10$^{-1}$ |
| 7. | ElasticNet | 1.07 | 9.06 x10$^{-1}$ | 0.52 | 0.37 | 3.08 | 9.06 x10$^{-1}$ |
| 8. | LassoLars | 1.06 | 9.18 x10$^{-1}$ | 0.52 | 0.37 | 3.06 | 9.18 x10$^{-1}$ |
| 9. | Ridge | 0.84 | 7.43 x10$^{-1}$ | 0.50 | 0.29 | 2.42 | 7.44 x10$^{-1}$ |

*boldface represents statistically significant results (p-val, logrank p<0.05). MLR hyperparameters and evaluation statistics are provided in Table 5 in S1 File.

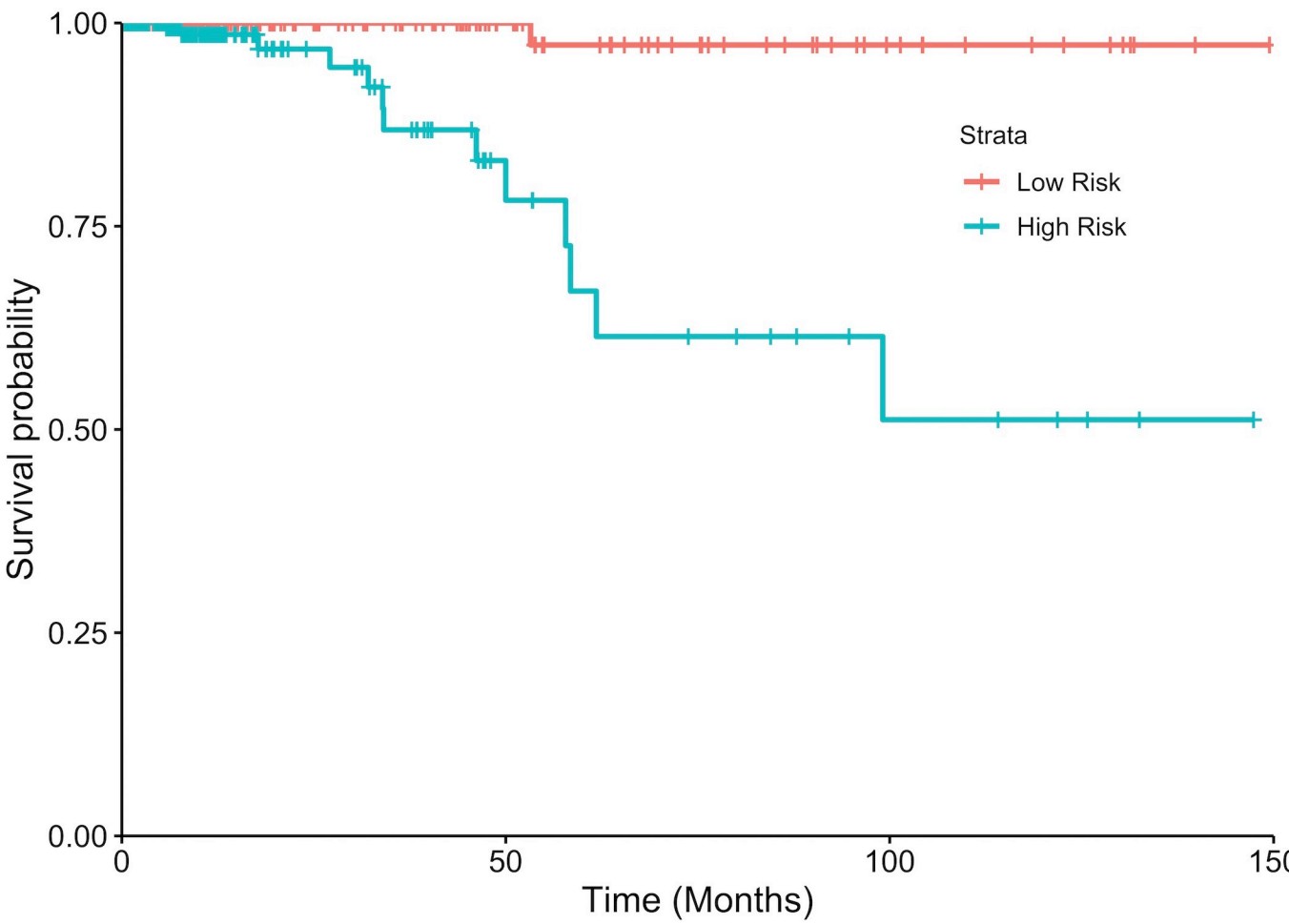

**Fig 1. KM plot showing risk stratification of PTC patients based on gene voting model.** Patients with greater than five 'high risk' labels in the 9-bit risk vector are assigned (blue) as High Risk (HR = 41.59, p = $3.36 \times 10^{-4}$, C = 0.84, logrank-p = $3.8 \times 10^{-8}$) while others were assigned as Low Risk (red).

## Multiple gene model sub-stratifies patients in clinico-pathological high-risk groups

Past studies indicate the role of certain clinico-pathological factors in PTC prognosis such as age, gender, ethnicity and tumour size [3, 4]. Thus, we performed a univariate analysis to assess the association of these factors with OS in our dataset. Table 3 shows the results of the univariate analysis. Patient age is seen to be the most significant factor in the PTC prognosis (HR = 48.65, C = 0.86), and is supported by numerous earlier studies [51]. The AJCC thyroid cancer staging also includes an age cut-off of 55 years to classify tumour stages [52], since patients with age<55y usually show a very good prognosis. However, we obtained the best stratification at the age cut-off of 60y which also corroborated with a recent study [53]. AJCC Tumour staging was seen to be the second-best risk predictor with HR = 9.23 and C = 0.76.

In order to evaluate the strength of the 9-gene based model, we sub-stratified the patients in the clinical high-risk subgroups i.e Age>60 and Stage III/IV patients. Fig 2 shows the sub-stratification by means of KM plots. A significant separation between the survival curves is seen, as denoted by logrank test's p-values. KM plots for other high-risk subgroups are provided in Fig 2 in S2 File.

**Table 3. Univariate analysis using clinico-pathological features.** Age is seen to be the most significant factor. In laterality, unilateral: right lobe, left lobe and isthmus.

| Factor | Strata | N | HR | p-val | C | %95 CI | | Logrank-p |
|---|---|---|---|---|---|---|---|---|
| **Age** | **>60 vs < = 60** | **505** | **48.65** | **1.85 x10$^{-4}$** | **0.86** | **6.35** | **372.82** | **7.32 x10$^{-9}$** |
| **Pathologic Stage** | **Stage III/IV vs I/II** | **503** | **9.23** | **6.61 x10$^{-4}$** | **0.76** | **2.57** | **33.17** | **1.05 x10$^{-4}$** |
| Tumour Focality | Unifocal vs Multifocal | 495 | 5.92 | 8.77 x10$^{-2}$ | 0.67 | 0.77 | 45.53 | 2.84 x10$^{-2}$ |
| Pathologic T stage | T3,T4 vs T1,T2 | 503 | 2.42 | 1.36 x10$^{-1}$ | 0.66 | 0.76 | 7.75 | 1.17 x10$^{-1}$ |
| Pathologic N stage | N1 vs N0 | 455 | 1.61 | 4.36 x10$^{-1}$ | 0.61 | 0.48 | 5.37 | 4.26 x10$^{-1}$ |
| Pathologic M stage | M1 vs M0 | 291 | 5.67 | 3.15 x10$^{-2}$ | 0.58 | 1.17 | 27.52 | 7.00 x10$^{-2}$ |
| Race | White vs Others | 413 | 2.20 | 4.49 x10$^{-1}$ | 0.56 | 0.29 | 16.81 | 3.96 x10$^{-1}$ |
| Gender | Male vs Female | 505 | 2.11 | 1.85 x10$^{-1}$ | 0.52 | 0.70 | 6.33 | 2.04 x10$^{-1}$ |
| Laterality | Bilateral vs Unilateral | 499 | 2.09 | 3.46 x10$^{-1}$ | 0.49 | 0.45 | 9.63 | 3.85 x10$^{-1}$ |
| Extrathyroidal extension | Yes vs No | 487 | 1.55 | 4.23 x10$^{-1}$ | 0.64 | 0.53 | 4.51 | 4.20 x10$^{-1}$ |
| Residual Tumour | R1,R2 vs R0 | 443 | 3.53 | 4.49 x10$^{-2}$ | 0.73 | 1.03 | 12.09 | 6.40 x10$^{-2}$ |

*boldface represents statistically significant results (p-val, logrank p<0.05).

## Hybrid voting model

After obtaining three prominent prognostic markers i.e. multiple gene voting model, patient age and AJCC stage, we performed a multivariate cox regression survival analysis. The analysis showed that patient age (HR = 13.3, p = 0.02) and gene voting model (HR = 13.3, p = 0.015) were independent covariates, while p-value corresponding to staging was insignificant as depicted by the forest plot in Fig 3A. Next, we developed a hybrid voting model by combining patient age with the 9-gene voting model for risk stratification purposes. The risk vector associated with each patient was thus now a 10-bit vector with 1 bit assigned to risk label due to age. Table 6 in S1 File shows results pertaining to stratification by hybrid models with different age cut-offs (45y-65y). We observed that the model performed best when the age cut-off was set at 65y (HR = 57.04, C = 0.88) as compared to 60y (HR = 54.82, C = 0.87). While the risk groups have a better separation in the former model, the 5 and 10-year survival is comparable in both models. High risk groups show a 40% 5-year survival and around 25% 10-year survival,

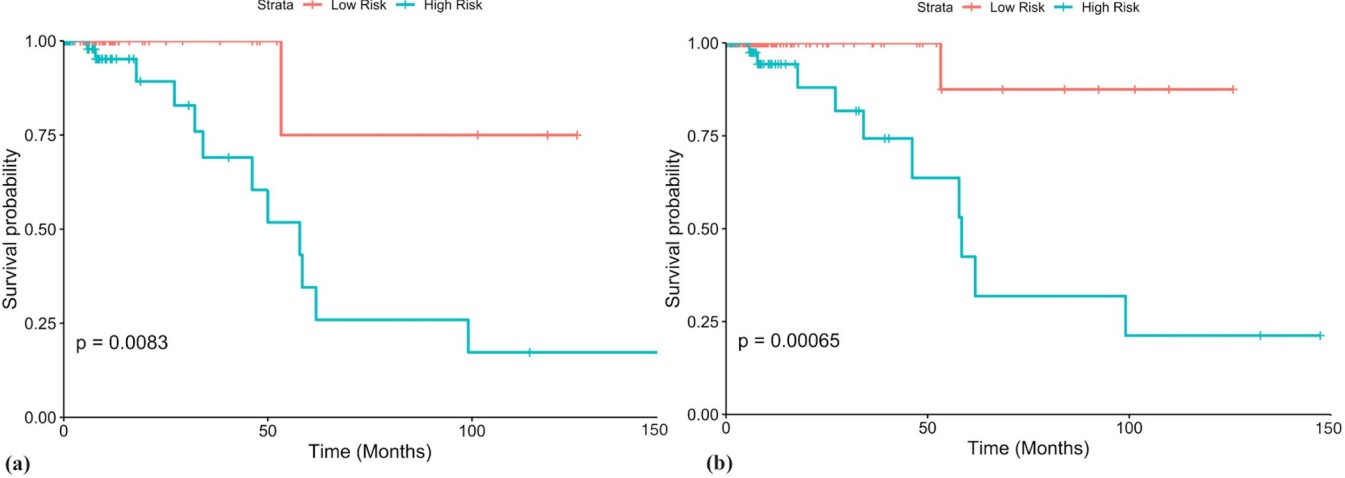

**Fig 2. Voting model sub-stratifies high risk groups.** (a) Patients with with age>60y (n = 113) were stratified into high and low risk groups with HR = 9.49, p = 3.08x10$^{-2}$ and C = 0.72. (b) Stage III/IV patients (n = 167) were stratified into high and low risk groups with HR = 15, p = 0.01 and C = 0.81. p-values from logrank tests are shown in the KM plots.

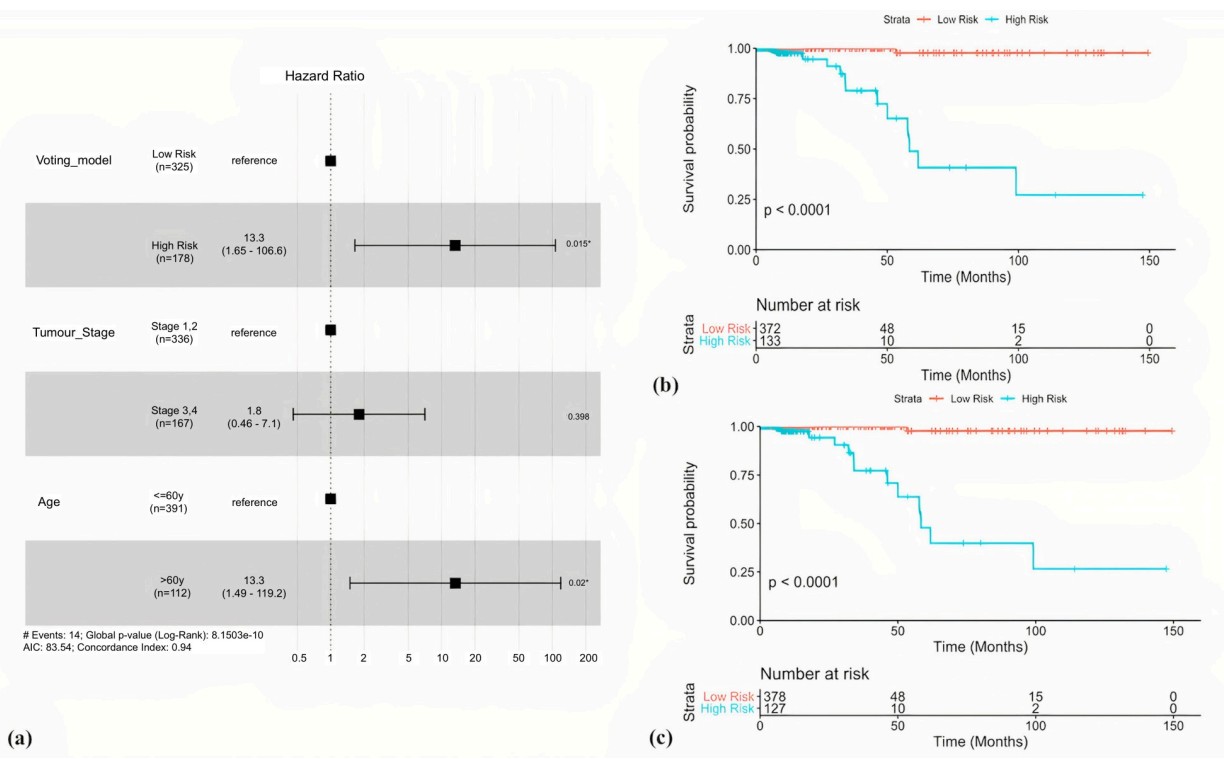

**Fig 3. Hybrid models for risk stratification.** (a) Multivariate analysis reveals Age (HR = 13.3, p = 0.02) and Voting model (HR = 13.3, p = 0.015) as two independent covariates, while tumour stage was found to be insignificant. (b) Risk stratification by hybrid model with age cut-off >60y (HR = 54.82, p = $1.18 \times 10^{-4}$, C = 0.87, %95CI: 7.14–420.90 and logrank-p = $2.3 \times 10^{-9}$). (b) (b) Risk stratification by hybrid model with age cut-off >65y (HR = 57.04, p~$10^{-4}$, C = 0.88, %95CI: 7.44–437.41 and logrank-p = $1.4 \times 10^{-9}$).

whereas, low risk groups have a 98% 5 and 10-year survival chance. Fig 3 shows the KM plots corresponding to both the hybrid models.

## Predictive validation

As implemented in [54] we performed a predictive assessment of our models using sub-samples of the complete dataset. Sampling sizes of 50%, 70% and 90% were chosen with 100 iterations each. HR and C index were evaluated for each iteration corresponding to the 9-gene voting model and hybrid models. Fig 4 shows the boxplots corresponding to the results. It is evident from the figure that the hybrid model with age cut-off of >65 years performs the best as compared to other models in terms of HR and C values. The median HR (27.03, 39.53, 50.33) and C (0.86, 0.87, 0.87) values for this model remain better than the other two models' despite of the sampling size. This method ensured that the risk stratification models were robust and performed well with random datasets of different sizes.

## Classification using hybrid model

In order to evaluate the classification performance of the above hybrid combination, we developed classification models. Firstly, we segregated patients into poor survival (negative data) and good survival (positive data) using an OS time cut-off. Secondly, we used package 'survivalROC' to calculate the true positive (TPR) and true negative rates (TNR). Here, a true positive prediction being the patient whose OS> cut-off time as well as who was in low-risk group according to hybrid model, while converse applies for a true negative prediction.

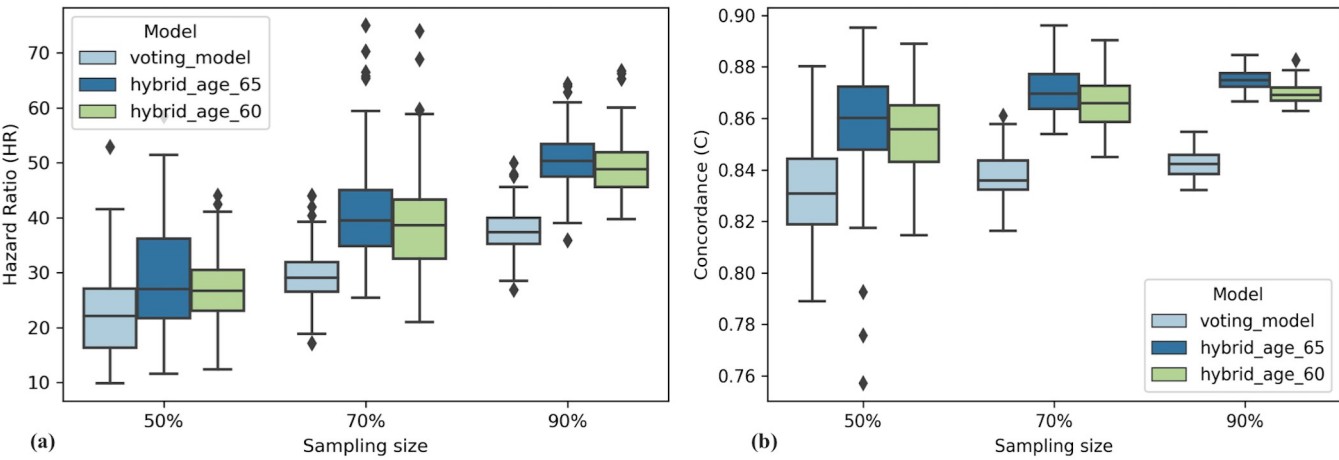

**Fig 4. Predictive validation of voting based model and hybrid models.** (a) Grouped boxplots corresponding to estimated Hazard Ratio (y-axis) for 100 iterations of data sampling (x-axis). (b) Similarly, estimation of Concordance index (y-axis) for different models using random sampling (x-axis).

Consequently, an AUROC value (Area under receiver operating characteristic curve) was calculated, which denoted the model's classification ability. Out of various time cut-offs used (2–10 years), the model was seen to perform best at the cut-off of 6 years. At this cut-off, a maximum AUROC value of 0.92 was obtained. The ROC curve is represented in Fig 5B.

## Validation of the prognostic gene signature

We compared the expression of these genes in normal patients (TCGA and GTEX normal samples) with cancer patients, with the help of GEPIA server [46]. Based on the results from GEPIA, it is found that the expression of *ANXA1*, *CLU*, *PSEN1*, *TNFRSF12A* and *GPX4* were up-regulated in THCA, while the expression of *TGFBR3* and *TIMP3* were down-regulated thus elucidating their role in PTC oncogenesis (Fig 6). While, the expression of *LEF1* and

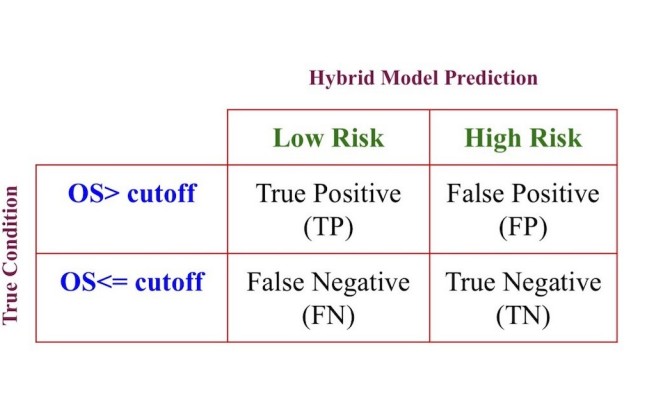

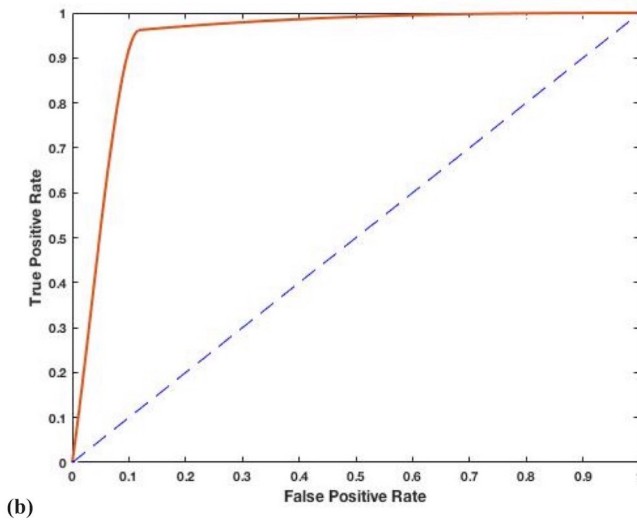

**Fig 5. Hybrid models for classification of PTC patients using OS.** (a) Terminology used for evaluation of confusion matrix. Initial risk labelling was done using an OS cut-off with patients having OS> cut-off labelled as positive or low risk and vice-versa for patients with OS≤cut-off. (b) ROC curve for hybrid model using age cut-off of 65y. AUROC of 0.92 was obtained.

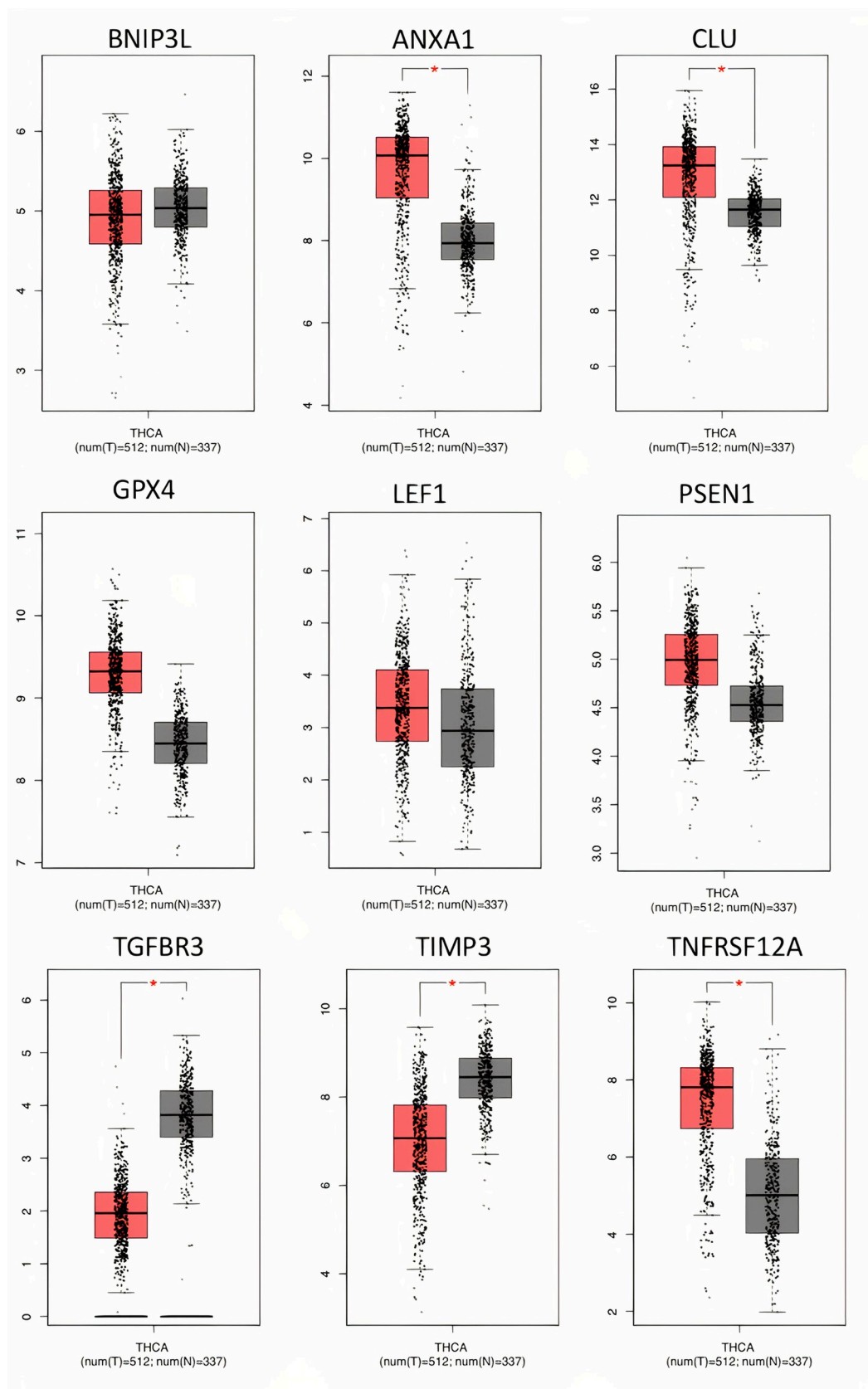

**Fig 6. Differential gene-expression analysis.** Boxplots representing the differential gene expression between normal and tumour samples on a log scale. GEPIA webserver was used to plot these by using TCGA THCA dataset. T: Tumour in red, N: Normal (TCGA, GTEX) in grey.

*BNIP3L* were found to have no significant difference. Thus, it indicates that the seven genes can be considered as differentially expressed genes (DEGs) in THCA compared to normal samples.

In addition, the protein expression patterns of the prognostic genes in THCA were performed using immunostaining data available at HPA (Fig 7) [55–61]. The results showed that *ANXA1* and *PSEN1* were highly expressed in THCA. Further medium expression of *GPX4* and *TNFRSF12A* were observed in THCA. Low expression of *CLU* was observed in THCA, but its expression was high at mRNA level. No expression of *TGFBR3* was observed in THCA. The expression of *LEF1* and *BNIP3L* was not detected in THCA tissues. These results validated our findings, except the candidate *CLU*. However, the expression of *TIMP3* was not recorded in HPA.

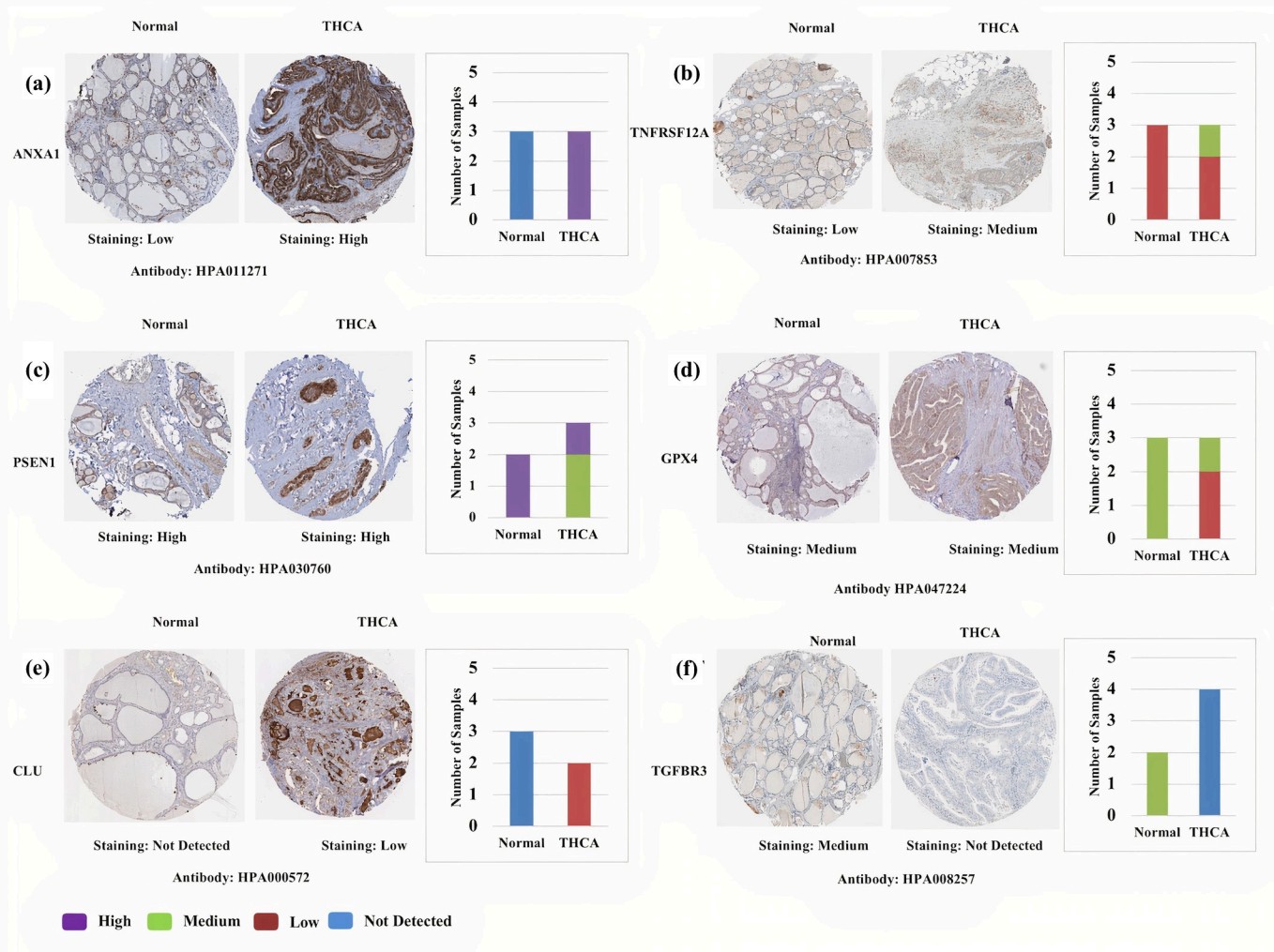

**Fig 7. The protein expression patterns of the prognostic genes from the Human Protein Atlas (HPA) database (proteinatlas.org).** (a) ANXA1, (b) PSEN1, (c) CLU, (d) TNFRSF12A, (e) GPX4, (f) TGFBR3. The staining intensity was annotated as High, Medium, Low and Not detected. The bar plots represent the number of samples with different staining intensity in HPA.

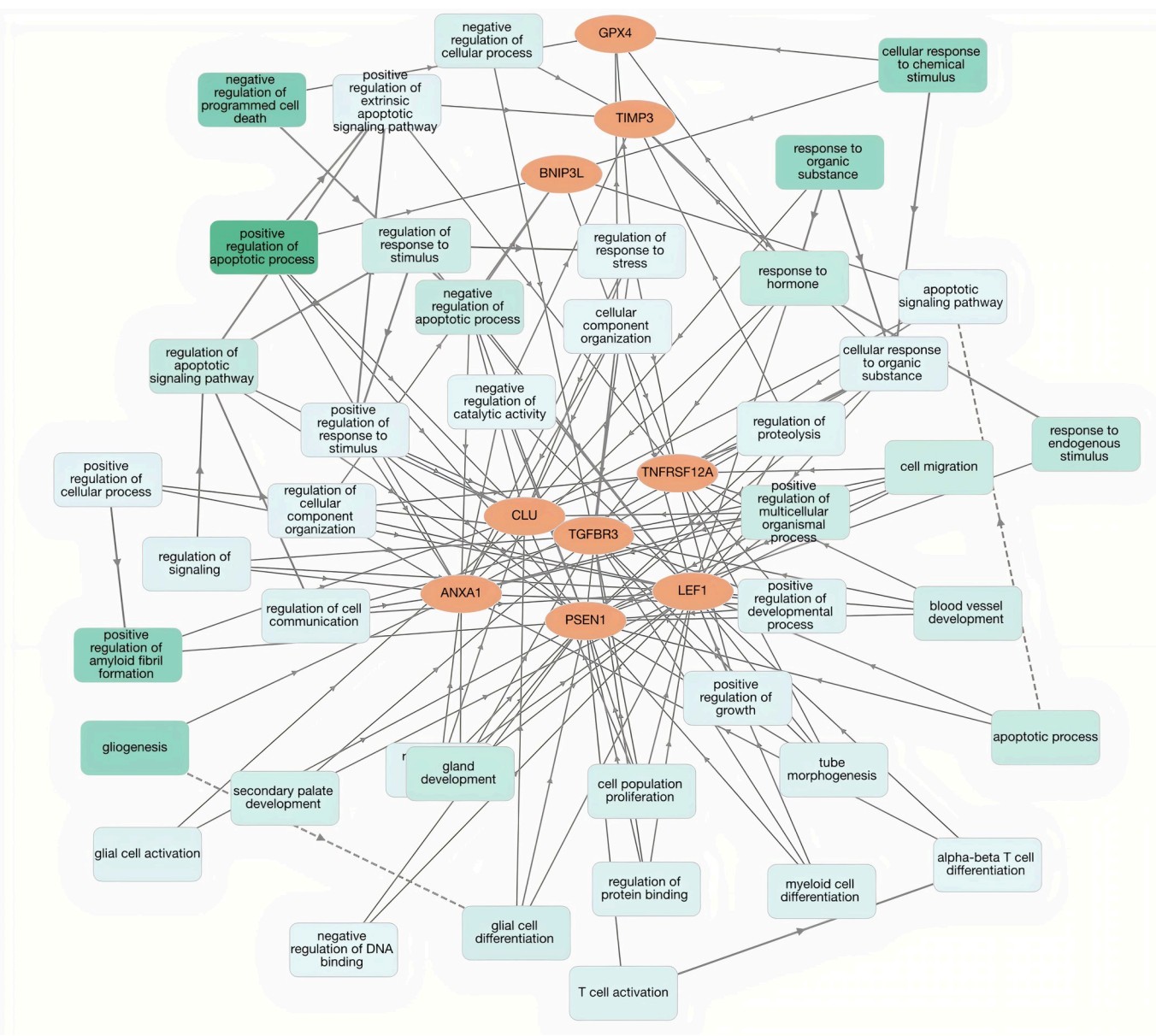

**Fig 8. Functional enrichment analysis.** The figure represents the significant biological process terms for the gene signatures. Orange color represents the prognostic genes; green color denotes significant biological process.

## Functional enrichment analysis

It is observed that the genes were significantly enriched in various biological process (BP) terms including positive regulation of apoptotic process, negative regulation of programmed cell death, gland development, positive regulation of amyloid fibril formation and cell migration (Fig 8).

## Screening of therapeutic drug molecules

Another major step after the identification of key genes whose altered expression is associated with PTC risk is the choice of therapy which can alter this situation. This requires selection of

small molecules which can induce or inhibit the gene expression of downregulated and upregulated genes in PTC. As implemented in [62], we found drug molecules which could reverse gene expression induced by PTC using the 'Cmap2 database' [63, 64]. A list of probe ids corresponding to upregulated genes (*TGFBR3*, *TIMP3*, *LEF1* and *BNIP3L*) and downregulated genes (*ANXA1*, *CLU*, *PSEN1*, *TNFRSF12A* and *GPX4)* was used as input to fetch small molecules ranked on the basis of p-values (results in Table 7 in S1 File). Top 2 negative and positively enriched molecules were Lomustine (enrichment = -0.908, p = 0.0001) and Deferoxamine (enrichment = 0.663, p = 0.0006). Lomustine is an alkylating nitrosourea compound which is already used in chemotherapy, especially in brain tumours, and has been associated with inducing apoptosis in past studies [65]. Deferoxamine (DFO) is an iron chelator which reduces iron content in cells. Various studies have confirmed that diminishing iron content inhibits tumor cell proliferation and induces apoptosis [66, 67]. Out of the various iron-chelators available, DFO is the most widely used iron-chelator and has shown to display these anti-tumor effects [68, 69].

## Discussion

Though PTC is known to have a very good prognosis; there still remains a decent proportion of patients with an abysmal prognosis. As a result of which, accurate risk assessment strategies are required for clinical decision making and therapeutic intervention. While conventional clinico-pathological factors such as age, stage, extrathyroidal spread and tumour size are significant in the risk stratification of PTC patients, they have their own limitations and are not that efficient. Thus, aided by the development in the high-throughput sequencing methods and availability of a huge amount of experimental data, various molecular prognostic markers have been proposed in the past [5–12, 14–16]. The understanding of the mechanistic roles of these molecules in the PTC carcinogenesis has initiated a further enquiry into other complicated molecular processes, which may be crucial in PTC progression and development. As uncovered in the past investigations, apoptosis in PTC is a multifaceted and multistep process. Apoptosis based biomarkers have also been proposed for many other cancers [17–19, 21, 22]. Despite the fact that the role of genes and their associated proteins such as *Fas/FasL*, *Bcl-2* family, *p53*, and others have been exhibited in PTC malignant growth, our comprehension of the collaboration between these molecules is still poor. The crosstalk that happens between numerous upstream signals and downstream effectors presents an extensive challenge to the ongoing investigation of apoptosis in PTC. Be that as it may, these complexities provide opportunities for disclosing novel prognostic biomarkers and therapeutic targets.

In the current study, we examined the genes involved in the apoptotic pathway and evaluated the prognostic potential of the expression of these genes in PTC. We employed a recent gene expression dataset, and found out that out of 165 genes, 9 genes were significantly associated with PTC prognosis. Out of these genes, *ANXA1* or annexin A1 expression has been shown to be associated with differentiation in PTC [70]. Western blotting experiments showed high levels of *ANXA1* in papillary thyroid carcinoma and follicular cells while undifferentiated thyroid carcinoma cells had low levels of *ANXA1* protein. *TGFBR3* gene was found to be differentially expressed between normal and PTC samples and was shown to be related to progression free interval [15]. The encoded *TGFBR3* protein is a membrane proteoglycan and is known to function as a co-receptor along with other *TGF-beta* receptor superfamily members. Reduced expression of the *TGFBR3* protein has also been observed in various other cancers. *CLU* protein is a secreted chaperone which has been previously suggested to be involved in apoptosis and tumour progression. Altered *CLU* expression has also been proposed as a biomarker for the assessment of indeterminate thyroid nodules [71]. *PSEN1* mutations have been

shown to be linked with MTC [72]. *TNFRSF12A* was linked to aging and thyroid cancer [73] and was also shown to be a PTC prognostic biomarker in yet another study [74]. *GPX4* is an essential seleno-protein shown to be associated with aging and cancer [75]. *TIMP3* levels were found to be associated with *BRAF* mutations in PTC [76]. *LEF1* expression was found to be up-regulated in PTC [77] and *BNIP3L-CDH6* interaction has been linked with defunct autophagy and epithelial to mesenchymal transition (EMT) in PTC [78]. We also evaluated the risk stratification performance of other genes suggested in past studies and showed that the 9 genes proposed in our study show better results. Moreover, out of 9 genes, 7 genes were found to be differentially expressed in THCA samples compared to normal samples, which are also supported by immunostaining results from HPA database. We also found potential drug-molecules which could be potentially used for PTC therapy and require future investigations. Lomustine and Deferoxamine were two such top molecules which are widely used in anti-cancer treatment due to their apoptosis inducing roles. Further, a multiple gene expression profile-based voting model was developed for these 9 genes. Apart from its superior performance in the complete dataset, this model was able to segregate high and low risk patients in clinically established high risk groups. We further gauged the performance of this multiple gene model against clinico-pathological factors, using a multivariate survival analysis. The analysis led to identification of 'Patient Age' as another independent significant factor, and thus a hybrid model utilizing the 9 gene expression profile and age was developed. This model further boosted the performance and provided better stratification. Further, Monte Carlo validation was performed to assess the robustness of this model. The model was also able to achieve an AUROC of 0.92 for classification of patients having more than 6 years overall survival with those having less than or equal to 6 years overall survival time. In conclusion, we identified key genes with a possible role in PTC pathogenesis and prognosis. While, this is supported by previous literature and explored in the current study as an in-silico analysis, it is subjected to further validation. Also, apart from their strong prognostic potential, as elucidated in this study, these genes could also be investigated further in the context of therapeutic targets in PTC and clinical decision making.

## Supporting information

**S1 File. The file contains additional information about the dataset, comparison studies and results pertaining to various risk stratification models.**
(XLSX)

**S2 File. The file contains Kaplan Meier plots for various models.**
(DOCX)

## Author Contributions

**Conceptualization:** Gajendra P. S. Raghava.

**Data curation:** Chakit Arora.

**Formal analysis:** Chakit Arora, Dilraj Kaur, Leimarembi Devi Naorem.

**Funding acquisition:** Gajendra P. S. Raghava.

**Investigation:** Chakit Arora, Dilraj Kaur, Gajendra P. S. Raghava.

**Methodology:** Chakit Arora.

**Project administration:** Gajendra P. S. Raghava.

**Resources:** Gajendra P. S. Raghava.

**Software:** Chakit Arora.

**Supervision:** Gajendra P. S. Raghava.

**Validation:** Chakit Arora.

**Visualization:** Chakit Arora, Dilraj Kaur, Leimarembi Devi Naorem.

**Writing – original draft:** Chakit Arora, Dilraj Kaur, Leimarembi Devi Naorem.

**Writing – review & editing:** Chakit Arora, Dilraj Kaur, Gajendra P. S. Raghava.

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
