## [Decision Letter · Decision Letter 0]

26 Aug 2021

PONE-D-21-18990

Prognostic Biomarkers for Predicting Papillary Thyroid Carcinoma Patients at High Risk Using Nine Genes of Apoptotic Pathway

PLOS ONE

Dear Dr. Raghava,

Thank you for submitting your manuscript to PLOS ONE. After careful consideration, we feel that it has merit but does not fully meet PLOS ONE’s publication criteria as it currently stands. Therefore, we invite you to submit a revised version of the manuscript that addresses the points raised during the review process.

We look forward to receiving your revised manuscript.

Kind regards,

Avaniyapuram Kannan Murugan, M.Phil., Ph.D.

Academic Editor

PLOS ONE

1. Please ensure that your manuscript meets PLOS ONE's style requirements, including those for file naming. The PLOS ONE style templates can be found at https://journals.plos.org/plosone/s/file?id=wjVg/PLOSOne_formatting_sample_main_body.pdf and https://journals.plos.org/plosone/s/file?id=ba62/PLOSOne_formatting_sample_title_authors_affiliations.pdf.

2. In order to ensure your study is reproducible, please provide the TCGA project dataset name(s) included in your study, and/or state the specific inclusion criteria used to select patient samples and the dates of your search

“Authors are thankful to Indraprastha Institute of Information Technology (IIIT-Delhi) and University Grants Commission (UGC, India) for financial support and fellowships.”

4. We note that Figure 7 in your submission contain copyrighted images. All PLOS content is published under the Creative Commons Attribution License (CC BY 4.0), which means that the manuscript, images, and Supporting Information files will be freely available online, and any third party is permitted to access, download, copy, distribute, and use these materials in any way, even commercially, with proper attribution. For more information, see our copyright guidelines: http://journals.plos.org/plosone/s/licenses-and-copyright.

a. You may seek permission from the original copyright holder of Figure 7 to publish the content specifically under the CC BY 4.0 license.

Additional Editor Comments:

As such the study is interesting and well-studied. Both the reviewers recommended for publication. Authors are asked to address the reviewer comments.  

Reviewers' comments:

Reviewer's Responses to Questions

**Comments to the Author**

1. Is the manuscript technically sound, and do the data support the conclusions?

Reviewer #1: Yes

Reviewer #2: Yes

2. Has the statistical analysis been performed appropriately and rigorously? 

Reviewer #1: Yes

Reviewer #2: Yes

3. Have the authors made all data underlying the findings in their manuscript fully available?

Reviewer #1: Yes

Reviewer #2: Yes

4. Is the manuscript presented in an intelligible fashion and written in standard English?

Reviewer #1: Yes

Reviewer #2: Yes

5. Review Comments to the Author

Reviewer #1: Manuscript entitled “Prognostic biomarkers for predicting papillary thyroid carcinoma patients at high risk using nine genes of apoptotic pathway” is a well-written manuscript with novelty. Authors have selected the various methods to obtain the data. The methods they used are appropriate and the data they presented show a clear support of their conclusion. Manuscript can be acceptable after proof reading for minor typos.

Reviewer #2: This is an interesting study and you have collected good datasets. The paper is well written and structured.

Line 1: Aberrant expression of apoptotic genes has been associated with papillary thyroid carcinoma. Have been is correct.

Line 28: Can you detail the the sentence, risk prediction models?

Line 86: Apoptosis is the process for eliminating cells in multicellular organisms? Can you add on additional points?

6. PLOS authors have the option to publish the peer review history of their article (what does this mean?). If published, this will include your full peer review and any attached files.

Reviewer #1: No

Reviewer #2: No

---

## [Author Response · Author response to Decision Letter 0]

10 Sep 2021

Reviewer #1: Manuscript entitled “Prognostic biomarkers for predicting papillary thyroid carcinoma patients at high risk using nine genes of apoptotic pathway” is a well-written manuscript with novelty. Authors have selected the various methods to obtain the data. The methods they used are appropriate and the data they presented show a clear support of their conclusion. Manuscript can be acceptable after proof reading for minor typos.

Response: We thank the reviewer for recommending our work. As suggested, we have re-checked the manuscript for typos and made the corrections.

Reviewer #2: This is an interesting study and you have collected good datasets. The paper is well written and structured.

Line 1: Aberrant expression of apoptotic genes has been associated with papillary thyroid carcinoma. Have been is correct.

Line 28: Can you detail the the sentence, risk prediction models?

Line 86: Apoptosis is the process for eliminating cells in multicellular organisms? Can you add on additional points?

Response: We appreciate the provided comments. We have now made the suggested correction and provided additional details for Line 28 and Line 86.

---

## [Editor Report · Decision Letter 1]

20 Sep 2021

PONE-D-21-18990R1Prognostic biomarkers for predicting papillary thyroid carcinoma patients at high risk using nine genes of apoptotic pathwayPLOS ONE

Dear Dr. Raghava,

Thank you for submitting your manuscript to PLOS ONE. I am pleased to inform you that your manuscript is acceptable for publication provided that you could address the minor issue in the Introduction part (kindly see Additional Editor Comments). Therefore, we invite you to submit a revised version of the manuscript that addresses the points raised during the review process.

We look forward to receiving your revised manuscript.

Kind regards,

Avaniyapuram Kannan Murugan, M.Phil., Ph.D.

Academic Editor

PLOS ONE

Journal Requirements:

Additional Editor Comments:Change the following line 33-35 ---It can be categorized into four major subtypes: i) papillary thyroid carcinoma (PTC), ii) follicular thyroid carcinoma (FTC), iii) medullary thyroid carcinoma (MTC), and 35 iv) anaplastic thyroid carcinoma (ATC).--- as exactly rewritten below and do not include MTC. "Thyroid cancer developed from follicular cells can be mainly categorized into papillary (PTC), follicular (FTC), and anaplastic thyroid cancer (ATC)."

---

## [Author Response · Author response to Decision Letter 1]

8 Oct 2021

Additional Editor Comments

1. Change the following line 33-35 ---It can be categorized into four major subtypes: i) papillary thyroid carcinoma (PTC), ii) follicular thyroid carcinoma (FTC), iii) medullary thyroid carcinoma (MTC), and 35 iv) anaplastic thyroid carcinoma (ATC).--- as exactly rewritten below and do not include MTC.

"Thyroid cancer developed from follicular cells can be mainly categorized into papillary (PTC), follicular (FTC), and anaplastic thyroid cancer (ATC)."

Response: Thank you. We have re-phrased the sentence as suggested by the Editor.

---

## [Editor Report · Decision Letter 2]

21 Oct 2021

Prognostic biomarkers for predicting papillary thyroid carcinoma patients at high risk using nine genes of apoptotic pathway

PONE-D-21-18990R2

Dear Dr. Raghava,

We’re pleased to inform you that your manuscript has been judged scientifically suitable for publication and will be formally accepted for publication once it meets all outstanding technical requirements.

Kind regards,

Avaniyapuram Kannan Murugan, M.Phil., Ph.D.

Academic Editor

PLOS ONE

---

## [Editor Report · Acceptance letter]

3 Nov 2021

PONE-D-21-18990R2 

Prognostic biomarkers for predicting papillary thyroid carcinoma patients at high risk using nine genes of apoptotic pathway 

Dear Dr. Raghava:

I'm pleased to inform you that your manuscript has been deemed suitable for publication in PLOS ONE. Congratulations! Your manuscript is now with our production department. 

Kind regards, 

on behalf of

Dr. Avaniyapuram Kannan Murugan 

Academic Editor

PLOS ONE